# Development and Characterization of ^99m^Tc-scFvD2B as a Potential Radiopharmaceutical for SPECT Imaging of Prostate Cancer

**DOI:** 10.3390/ijms25010492

**Published:** 2023-12-29

**Authors:** Cristina Bolzati, Carolina Gobbi, Guillermina Ferro-Flores, Sofia Turato, Blanca Ocampo-Garcia, Debora Carpanese, Cristina Marzano, Barbara Spolaore, Giulio Fracasso, Antonio Rosato, Laura Meléndez-Alafort

**Affiliations:** 1Institute of Condensed Matter Chemistry and Energy Technologies, Consiglio Nazionale delle Ricerche, Corso Stati Uniti 4, 35127 Padova, Italy; cristina.bolzati@cnr.it (C.B.); carolina.gobbi@icmate.cnr.it (C.G.); 2Laboratorio Nacional de Investigación y Desarrollo de Radiofármacos, Instituto Nacional de Investigaciones Nucleares, Carretera México-Toluca S/N. La Marquesa, Ocoyoacac 52750, Mexico; guillermina.ferro@inin.gob.mx (G.F.-F.); blanca.ocampo@inin.gob.mx (B.O.-G.); 3Istituto Oncologico Veneto IOV-IRCCS, Via Gattamelata 64, 35138 Padova, Italy; sofiaturato11@gmail.com (S.T.); antonio.rosato@unipd.it (A.R.); 4Dipartimento di Scienze del Farmaco, Università degli Studi di Padova, Via Marzolo 5, 35131 Padova, Italy; cristina.marzano@unipd.it (C.M.); barbara.spolaore@unipd.it (B.S.); 5Dipartimento di Scienze Biomediche, Università degli Studi di Padova, Viale G. Colombo 3, 35131 Padova, Italy; 6Dipartimento di Scienze Chirurgiche Oncologiche e Gastroenterologiche, Università degli Studi di Padova, Via Gattamelata 64, 35138 Padova, Italy

**Keywords:** ^99m^Tc-HYNIC-scFvD2B, ^99m^Tc-HscFvD2B, prostate cancer imaging, technetium, single chain variable fragment, mAb D2B, PSMA

## Abstract

Previously, we demonstrated that the ^177^Lu-labeled single-chain variable fragment of an anti-prostate-specific membrane antigen (PSMA) IgG D2B antibody (scFvD2B) showed higher prostate cancer (PCa) cell uptake and tumor radiation doses compared to ^177^Lu-labeled Glu-ureide-based PSMA inhibitory peptides. To obtain a ^99m^Tc-/^177^Lu-scFvD2B theranostic pair, this research aimed to synthesize and biochemically characterize a novel ^99m^Tc-scFvD2B radiotracer. The scFvD2B-Tag and scFvD2B antibody fragments were produced and purified. Then, two HYNIC derivatives, HYNIC-Gly-Gly-Cys-NH_2_ (HYNIC-GGC) and succinimidyl-HYNIC (S-HYNIC), were used to conjugate the scFvD2B-Tag and scFvD2B isoforms, respectively. Subsequently, chemical characterization, immunoreactivity tests (affinity and specificity), radiochemical purity tests, stability tests in human serum, cellular uptake and internalization in LNCaP(+), PC3-PIP(++) or PC3(−) PCa cells of the resulting unlabeled HYNIC-scFvD2B conjugates (HscFv) and ^99m^Tc-HscFv agents were performed. The results showed that incorporating HYNIC as a chelator did not affect the affinity, specificity or stability of scFvD2B. After purification, the radiochemical purity of ^99m^Tc-HscFv radiotracers was greater than 95%. A two-sample *t*-test of ^99m^Tc-HscFv1 and ^99m^Tc-HscFv1 uptake in PC3-PIP vs. PC3 showed a *p*-value < 0.001, indicating that the PSMA receptor interaction of ^99m^Tc-HscFv agents was statistically significantly higher in PSMA-positive cells than in the negative controls. In conclusion, the results of this research warrant further preclinical studies to determine whether the in vivo pharmacokinetics and tumor uptake of ^99m^Tc-HscFv still offer sufficient advantages over HYNIC-conjugated peptides to be considered for SPECT/PSMA imaging.

## 1. Introduction

Prostate-specific membrane antigen (PSMA) has become the most important biomarker for prostate cancer diagnosis and therapy, as PSMA has been found to be overexpressed in 95% of PCa tumors, with increasing expression levels in higher-grade and metastatic lesions [1]. Among all the PSMA target molecules, the radiolabeled urea-based PSMA inhibitors are the most studied agents due to their high specific binding to the PSMA and their rapid clearance from non-target tissues [2]. PSMA diagnostic agents are mostly labeled with ^68^Ga to obtain positron emission tomography (PET) images with a high sensitivity and spatial resolution [3]. ^68^Ga-PSMA-11 (Glu-urea-Lys(Ahx), using N,N′-bis [2-hydroxy-5-(carboxyethyl)benzyl] ethylenediamine-N,N′-diacetic acid) as a ^68^Ga chelator, has been shown to be the best complex for the imaging of PCa and metastatic lesions due to its high binding affinity, thermodynamic stability and easy preparation compared to the ^68^Ga-DOTA conjugate variants, known as ^68^Ga-PSMA-617 and ^68^Ga-PSMA-I&T [4,5,6]. However, the cost of a ^68^Ge/^68^Ga generator and the amount of radionuclide that can be obtained in a single elution, sufficient only for a few patients, can limit the application of ^68^Ga radiopharmaceuticals in clinical settings [4]. In contrast, the less-expensive ^99^Mo/^99m^Tc generator produces enough ^99m^Tc to treat many patients daily. In addition, ^99m^Tc is considered the workhorse of nuclear medicine due to its ideal emission properties for single-photon emission computed tomography (SPECT) imaging [7]. Consequently, preclinical and clinical studies of ^99m^Tc-PSMA complexes have recently been performed to evaluate their potential as radiopharmaceuticals for PCa diagnosis using PSMA-derived inhibitors radiolabeled with [^99m^Tc(CO)_3_]^+^ [8,9,10], ^99m^Tc(V) Oxo Core [11,12], ^99m^Tc-HYNIC [13,14,15,16,17] to produce a wide range of PSMA imaging agents [18].

Another strategy pursued in the development of PCa diagnostic agents exploits antibody-based constructs, as antibodies generally have a higher affinity for their targets than small peptides [19]. In fact, the monoclonal antibody (mAb) ^111^In-capromab pendetide (ProstaScint^®^ Cytogen, Princeton, NJ, USA; murine origin) was the first Ab-based radiopharmaceutical approved by the Food and Drug Administration (FDA) for PCa diagnosis [20,21,22,23]. However, because ProstaScint^®^ recognizes an intracellular epitope, its binding sites are exposed only after apoptosis or necrosis processes. To overcome this problem, a humanized mAb called J591, which recognizes an extracellular epitope of PSMA, was developed, showing promising results in clinical diagnosis [24,25,26]. Unfortunately, as a whole antibody, it exhibits slow clearance from non-target tissues and a maximum tumor uptake from 6 to 7 days after injection [27].

Recently, Colombatti et al. generated the IgG-D2B mAb against an external epitope of human PSMA using conventional hybridoma technology. The tumor uptake of ^111^In-labeled IgG-D2B was shown to be higher than that of the commercial ProstaScint^®^ but suffered from the same molecular size limitations [28]. Therefore, to circumvent this problem, a single-chain variable fragment of the IgG-D2B antibody (termed scFvD2B) was generated [29]. It has been reported that small antibody fragments can access binding sites more uniformly and penetrate solid tumors more homogeneously and efficiently than full-length antibodies [30]. In vivo imaging of scFvD2B radiolabeled with different radionuclides (^123^I, ^111^In and ^177^Lu) confirmed the relatively fast and high antigen-positive tumor uptake (from 3 h post-injection), with a short circulatory half-life and rapid clearance from most of the non-target tissues, allowing labeling with radionuclides of relatively short physical half-life (t_1/2_) such as ^99m^Tc (t_1/2_ = 6 h) [31,32,33]. Moreover, due to its larger size compared to PSMA-derived peptides, scFvD2B can be conjugated to different chelating agents without drastically affecting its binding affinity and pharmacokinetic properties [29,32].

Among the bifunctional chelating agents used for ^99m^Tc radiolabeling, HYNIC is currently one of the most widely used to label biomolecules in the clinic because it can be incorporated into amino acid building blocks and presents a diverse choice of co-ligands [34].

In this study, scFvD2B was conjugated to one of the HYNIC derivatives (HYNIC-GGC and S-HYNIC) and labeled with ^99m^Tc. Subsequently, the stability, immunoreactivity, cellular uptake and internalization of the resulting ^99m^Tc-HscFv agents were evaluated to establish their potential as radiopharmaceuticals for SPECT imaging of PCa.

## 2. Results

The scFvD2B-Tag and scFvD2B antibody fragments were produced in a prokaryotic system in our laboratory and in a eukaryotic system by ExcellGene, respectively. They were purified using affinity chromatography, as reported by Frigerio et al. [14]. The only difference between the two antibody fragments is that scFvD2B-Tag contains a sequence of six histidyl residues attached to the C-terminus of the protein, which allows the easy purification of scFvD2B [18]. Two HYNIC derivatives, HYNIC-Gly-Gly-Cys-NH_2_ (HYNIC-GGC) and succinimidyl-HYNIC (S-HYNIC), were used to conjugate scFvD2B-Tag and scFvD2B, respectively (Figure 1).

### 2.1. Conjugation of HYNIC-GGC-NH_2_ Peptide to scFvD2B-Tag

The first conjugation reaction was carried out with the activation of the accessible carboxylic groups of scFvD2B-Tag with S-NHS (N-hydroxysuccinimide) and EDC (N-(3-Dimethylaminopropyl)-N′-ethylcarbodiimide hydrochloride) reagents to form an active NHS ester (Figure 1). In principle, the NHS ester obtained could react with the thiol group of cysteine, the terminal amide of the HYNIC-GGC-NH_2_ peptide and the hydrazine group of HYNIC moiety. However, under these reaction conditions, the thiol group has a greater ability to form stable thioester bonds (-S-C=O) than the nitrogen atom of the cysteine amide group, where the formation of an O=C-NH-C=O bond is known to be chemically unstable and unlikely [35]. The scFvD2B that could be conjugated through the HYNIC hydrazine group will not be a problem for imaging studies because it is not chemically available for coordination with ^99m^Tc.

At the end of the reaction, the product was purified with a Vivaspin^®^ centrifugal concentrator (MWCO 5 kDa) using PBS (0.1 M, pH 7) as a washing buffer to remove the unconjugated HYNIC-GGC-NH_2_. The purified HYNIC-GGC-scFvD2B-Tag (HscFv1) was analyzed using size exclusion high-performance liquid chromatography (SE-HPLC) to determine its grade of purity and using matrix-assisted laser desorption ionization mass spectroscopy (MALDI-MS) to confirm the conjugation.

SE-HPLC chromatograms of HscFv1 and unconjugated scFvD2B-Tag obtained using method A (see experimental section) showed a main peak with a similar retention time, indicating that the antibody fragment was not degraded or aggregated (Figure 2A). MALDI-MS analysis was used to confirm the production of HscFv1 (Figure 2B) and calculate the number of HYNIC-GGC conjugated to each scFvD2B-Tag. The number was obtained by dividing the mass difference between the main peaks of conjugated (29,999 Da) and unconjugated (29,098 Da) compounds by the mass of HYNIC-GGC (369.12 Da). The results showed an average of two to three HYNIC-GGC molecules conjugated to each scFvD2B-Tag.

Analysis of the unconjugated fragment also showed a small peak with a molecular weight of about 58,000 Da, which is probably an aggregate of scFvD2B-Tag (Figure 2B). The aggregated conjugate showed an average of four HYNIC-GGC, as demonstrated by the MALDI-MS analysis of HscFv1. Of note, the conjugation reaction does not increase the amount of aggregate. Moreover, before radiolabeling, the HscFv1 was purified from the aggregates.

### 2.2. Assessment of the Affinity of HscFv1 for PSMA

An ELISA assay and displacing experiments using flow cytometry were performed to evaluate the affinity and specificity of scFvD2B-Tag and HscFv1 for the PSMA.

The ELISA test showed that both biomolecules exhibited a very similar scFv-antigen dissociation constant (K_D_), 7.4 and 9.1 nM for HscFv1 and scFvD2B-Tag, respectively, suggesting that the conjugation procedure did not significantly reduce the affinity of the scFv for the receptors (i.e., binding capability in the nanomolar range). Moreover, the binding to the bovine serum albumin (BSA) did not increase after HYNYC-GGC conjugation, indicating that this procedure does not modify the specificity associated with the molecular recognition (Figure 3).

Displacing experiments on LNCaP PSMA+ cells were also carried out to confirm the maintenance of the binding specificity of scFvD2B-Tag and its derivative HscFv1. Then, the binding of Ab D2B to LNCaP cells was analyzed in the presence of different concentrations (i.e., 500×, 25×, or 1× molar excess) of scFvD2B-Tag or HscFv1. As depicted in Figure 4, when LNCaP cells were co-incubated with a decreased molar excess of scFvD2B-Tag (Figure 4C–E) or HscFv1 (Figure 4G–I) and a fixed amount of the parental whole Ab D2B, the percentage of stained cells detected by flow cytometry increased from 4.3% to 99.8% and from 9.5% to 99.7%, respectively. The results of displacing clearly confirmed that HscFv1 preserved the target specificity of unconjugated scFvD2B-Tag (Figure 4).

### 2.3. Radiolabeling of HscFv1 Conjugate and Stability Test of ^99m^TcHscFv1

^99m^Tc-HscFv1 (^99m^Tc-HYNIC-GGC-scFvD2B-Tag) was obtained after a 60 min incubation of HscFv1 with [^99m^Tc]-pertechnetate at 37 °C and neutral pH in the presence of two coligands (EDDA and tricine) to complete the metal coordination sphere. The radiolabeling yield (RLY) was assessed using SE-HPLC chromatography (using method A), as reported in the literature for other single-chain antibody fragments conjugated to HYNIC [36]. HPLC-chromatograms showed a main peak (RLY > 99%) with an Rt overlapping that of unlabeled HscFv1, and no presence of pertechnetate was observed (Figure 5A). Furthermore, 98% of radioactivity recovery after HPLC analysis demonstrated the non-formation of ^99m^Tc-colloids. Under these analysis conditions, ^99m^Tc-HYNIC-GGC showed the same retention time as the radiolabeled HscFv1. Therefore, the product was also analyzed using C4 reverse-phase HPLC (method B) in order to identify whether the obtained peak was ^99m^Tc-HscFv1 or ^99m^Tc-HYNIC-GGC. The results showed that ^99m^Tc-HscFv1 was less than 5%, so most of the radioactivity was identify as the ^99m^Tc-HYNIC-GGC species (Figure 5B). Before radiolabeling, the HscFv1 conjugate was purified from the unconjugated HYNIC-GGC-NH_2_ peptide. Therefore, the results could be explained by the fact that probably the reaction conditions are responsible for the hydrolysis of the thioester group with consequent displacement of the HYNIC-GGC peptide from HscFv1.

^99m^Tc-HscFv1 was purified using the size exclusion column NAP5, obtaining a radioconstruct with a radiochemical purity (RCP) greater than 95%, which was used to perform the stability test and the cellular studies.

### 2.4. Conjugation of S-HYNIC to scFvD2B

Despite the fact that the ^99m^Tc-HscFv1 conjugate preserved good stability and affinity for PSMA, the His-Tag sequence could have a negative impact on the overall protein structure and function, as well as introduce a potential immunogenic site, so for most pharmaceutical-grade proteins the natural structure is usually required [37,38]. Thus, we shifted our attention to the corresponding no-tag scFv to also have a direct comparison between the two fragments.

To solve the problem of ^99m^Tc-HscFv1 hydrolysis, a new conjugate was synthesized using the commercially available succinimidyl-HYNIC hydrochloride (S-HYNIC). Two solutions of scFvD2B with concentrations of 5 and 2 mg/mL were tested (Figure 2).

The first attempt was made using the most concentrated solution of scFvD2B (5 mg/mL) and a 1:20 protein/S-HYNIC molar ratio. The result of the reaction was a precipitate with the characteristic yellow colour of HYNIC-conjugates, not useful in radiopharmaceutical preparation. A clear S-HYNIC-scFvD2B (HscFv2) adduct solution was obtained when the concentration of scFvD2B was decreased to 2 mg/mL and the protein/S-HYNIC molar ratio to 1:10. The HscFv2 conjugate was purified as described above to remove the unconjugated S-HYNIC and analyzed using C_4_-RP-HPLC (method B) and liquid chromatography–mass spectrometry (LC–MS) to confirm the conjugation.

The C_4_-RP-HPLC chromatogram showed a single peak with a retention time similar to that of the unconjugated scFvD2B but a different shape, indicating the formation of at least one new product (Figure 6A). LC–MS analysis revealed the presence of unconjugated scFvD2B and four adducts with different numbers of HYNIC molecules because S-HYNIC reacts with the side-chain ε amino groups of various Lys amino acids present on the scFvD2B chain. Conjugates containing one or two HYNIC molecules are those with greater abundance (Figure 6B).

### 2.5. Assessment of the Affinity for PSMA of the HscFv2 Conjugate

The binding affinity and specificity of the HscFv2 conjugate were also assessed with an ELISA assay. As already observed for the HscFv1 derivative, no significant changes in binding capability and specificity were found when applying this second derivatization procedure. In fact, the scFv-PSMA K_D_ values of HscFv2 (4.7 nM) and scFvD2B (5.0 nM) are very similar and slightly lower than those of the corresponding scFv-Tag derivatives. Moreover, binding curves to the unrelated protein BSA did not show any increase in the signal (Figure 7).

### 2.6. Radiolabeling of HscFv2 Conjugate

HscFv2 (S-HYNIC-scFvD2B) was labeled under mild conditions with ^99m^Tc-pertechnetate as described for HscFv1 in the presence of EDDA and tricine as coligands. The C_4_-RP-HPLC radiochromatogram of ^99m^Tc-HscFv2 showed an RLY of 45% (Figure 8); this was then purified with a size exclusion NAP5 column. Upon purification, the RCP of ^99m^Tc-HscFv2 was higher than 95%.

### 2.7. Stability Test of ^99m^TcHscFv1 and ^99m^TcHscFv2

The stability of purified ^99m^Tc-HscFv1 and ^99m^Tc-HscFv2 was evaluated with HPLC after a 1:10 dilution in phosphate-buffered saline (PBS), or saline solution (SS). The C4-RP HPLC chromatograms of ^99m^Tc-HscFv1 obtained after incubation at 37 °C over time (from 1 to 4 h) showed that a small peak (<2%) appeared after 4 h (Figure 9A). Dilutions of ^99m^Tc-HscFv2 in PBS, SS and 1:10 in human serum were analyzed for a more extended time (24 h) with incubation at 37 °C. The results showed that no peaks appeared (Figure 9B), except a minimal amount of pertechnetate one hour after dilution in human serum, but it does not increase over time (Figure 9C). These results confirmed the high stability of ^99m^Tc-HscFv2.

### 2.8. Assessment of In Vitro Uptake and Internalization

Before starting cell studies, we confirmed that the RCP of both labeled conjugates was greater than 95%. Cell uptake and internalization were assessed with incubation of 5 μL of pure ^99m^Tc-HscFv1 or ^99m^Tc-HscFv2 (5 µCi/5 µL) with 1 mL of RPMI cell culture media containing 1 × 10^6^ of PSMA+/− cells (i.e., LNCaP(+), PC3-PIP(++) or PC3(−)) at 37 °C for 1.5 h (in triplicate).

The results of the Shapiro–Wilk normality test confirmed that the data distribution of uptake and internalization came from a normal distribution (Appendix A) (Table A1). PSMA uptake of ^99m^Tc-HYNIC-scFvD2B agents was higher in PSMA-positive cells than in the negative controls. The two-sample t-test of ^99m^Tc-HscFv1 and ^99m^Tc-HscFv1 in PC3-PIP vs. PC3 showed *p*-values of 6.5 × 10^−^^19^ and 7.3 × 10^−^^8^, respectively (*p* < 0.001), indicating that the difference was statistically significant. Uptake and internalization in transduced PC3-PIP cells of purified ^99m^Tc-HscFv1 (13.9 ± 0.5 and 10.6 ± 1.1%) and ^99m^Tc-HscFv2 (12.7 ± 1.4 and 11.0 ± 1.5%) showed no statistical differences; the *p*-values obtained in a *t*-test analysis of uptake and internalization data of ^99m^Tc-HscFv1 vs ^99m^Tc-HscFv2 were 0.064 and 0.601, respectively. In both cases, blocking the cell receptors with an excess of cold scFvD2B significantly decreased the uptake to 1.4 ± 0.1% and 4.3 ± 0.1% for ^99m^Tc-HscFv1 and ^99m^Tc-HscFv2, respectively, confirming the site-specific interaction (Figure 10).

## 3. Discussion

This study evaluated the feasibility of using technetium-labeled scFvD2B as a theranostic pair for ^177^Lu-labeled scFvD2B. ^177^Lu-scFvD2B is currently under preclinical investigation because it has shown increased prostate cancer cell uptake, internalization and tumor radiation dose as compared to the Glu-ureido-based PSMA inhibitor peptides labeled with ^177^Lu [31,33]. Different types of chelators can be used for scFvD2B because, unlike PSMA inhibitor peptides, the pharmacokinetic properties (including cellular internalization and biodistribution of scFvD2B), have been shown not to be significantly affected by the chelator [31,32]. The decision to label the scFvD2B with ^99m^Tc was based on the fact that it has a slower tumor uptake (3 h) than PSMA peptides [33] and is therefore not a good candidate for labeling with a very short half-life radionuclide such as ^68^Ga. Moreover, despite the higher spatial resolution and sensitivity of ^68^Ga-PSMA agents, some recently developed ^99m^Tc-PSMAs have proven to be a cost-effective alternative to ^68^Ga tracers due to the lower cost and greater availability of ^99m^Tc, as well as the presence of SPECT in most hospitals worldwide [18]. HYNIC was selected as the ^99m^Tc chelating agent because it has been used to label high-efficiency PSMA peptides, using EDDA and Tricine to complete the coordination of technetium, under mild conditions [15,18]. In addition, HYNIC can be used to easily develop lyophilized kit formulations useful for clinical translation of radiopharmaceuticals such as ^99m^Tc-EDDA/HYNIC-iPSMA and PSMA-T4, which are currently in Phase I–II clinical trials [13,15].

In this work, two HYNIC-scFvD2B conjugates were developed using two different approaches: the first exploits the coupling of the thiol group of the Cys residue of the HYNIC-GGC peptide to the activated carboxyl groups of the scFvD2B-Tag; the second one exploits the conjugation of the conventional S-HYNIC to the **ε**-amino group of lysine residues on the scFvD2B. Chemical characterization of the two derivatives showed that HscFv1 containsan average of two to three HYNIC-GGC molecules conjugated to each scFvD2B-Tag, whereas the HscFv2 conjugates with the highest abundance were those containing one or two HYNIC molecules. However, both showed a comparable binding affinity; the K_D_ values obtained were 7.4 nM and 4.7 nM for HscFv1 and HscFv2, respectively. Their affinity was slightly weaker than that of ^99m^Tc-EDDA/HYNIC-iPSMA (3.11 nM), very similar to that of PSMA-T4 (5.4 nM) and higher than that reported for the three HYNIC-Lys-urea-Aad-derived molecules (from 8.96 to 11.6 nM) by Lu et al. [15,39].

Nevertheless, HscFv1 was found to be unstable under radiolabeling conditions and a very low amount of the corresponding ^99m^Tc-HscFv1 was obtained, as shown by the combination of the two methods of HPLC analysis, where more than 90% of the eluted radioactivity was identified as ^99m^Tc-HYNIC-GGC. However, the purified ^99m^Tc-HscFv1 proved to be stable long enough (4 h) to perform the diagnostic imaging and its stability was comparable to that reported by Maurin et al. for PSMA-T4 [15].

A more efficient radiosynthesis was attained for ^99m^Tc-HscFv2. The radioconstruct was prepared with an RLY of 45% reacting in mild conditions at 37 °C with a very small amount of protein (9.1–2.22 nmol). NAP 5 purification is required to obtain an RCP > 95%. It should be noted that the concentration of the construct is absolutely comparable to that used for HYNIC-PSMA inhibitor peptides for the preparation of ^99m^Tc-EDDA/HYNIC-iPSMA and HYNIC-ALUG [13,40], for which the quantitative formation (RLY > 98%) of the ^99m^Tc-tagged PSMA inhibitor is generated with harsher reaction conditions: heating at 100 °C for 15–20 min.

The in vitro stabilities of the pure ^99m^Tc-HscFv2 show that it is more stable than ^99m^Tc-HscFv1. In addition, the serum stability of ^99m^Tc-HscFv2 was similar to that of PSMA-T4 [15] and much higher than that of the three urea-based peptides labeled with ^99m^Tc using HYNIC reported by Mosayebnia et al. (less than 80% 4 h after dilution) [41].

Both ^99m^Tc-HscFv1 and ^99m^Tc-HscFv2 showed significant differences in radiopharmaceutical uptake in PSMA(+) and PSMA(−) cell lines and in the percentage of uptake and internalization between blocked and unblocked cells. The uptake in LNCaP cells of ^99m^Tc-HscFv2 (7.1 ± 0.7%) after 2 h incubation at 37 °C was higher than that reported for ^99m^Tc-EDDA/HYNIC-iPSMA (4.5 ± 0.8%) and ^99m^Tc-HYNIC-ALUG (1.2 ± 0.2%) [13,40]; however, the greatest difference was found in cell internalization, as only about 14% of the total uptake was internalized by ^99m^Tc-EDDA/HYNIC-iPSMA compared to the 80% of ^99m^Tc-HscFv2 uptake, which correspond to 0.63% and 5.6% of the radioactivity added to the cells, respectively.

It is worth noting that the overall findings show that scFvD2B-Tag retains the same stability and PSMA affinity properties as the corresponding native scFvD2B, which indicates that the presence of the His-Tag does not have a negative effect on the protein function. Nevertheless, modification with HYNIC-GGC peptide is not suitable for the development of a Tc-tagged construct due to the instability of the thioester group formed under the radiolabeling conditions. To overcome this issue, alternative conjugation methods and labeling approaches should be considered.

## 4. Materials and Methods

The succinimidyl-HYNIC (S-HYNIC) hydrochloride was obtained from ABX advanced biochemical compounds GmbH, and the HYNIC-GGC peptide (hydrazinonicotinyl-glycine-glycine-cysteine, MW 369.4) was obtained from Ontores (Ontores Biotechnology Co., Shangai, China) both with a high purity grade (>98%). The [^99m^Tc]-pertechnetate was obtained from a ^99^Mo/^99m^Tc UTK-generator (GE Healthcare, Chicago, IL, USA). scFvD2B (MW 27 kDa) and scFvD2B-Tag (29 kDa) were produced in a eukaryotic system (ExcellGene) and purified on a protein L-sepharose column (GE Healthcare) as previously described [16,17]. All the other reagents were purchased from Sigma-Aldrich and used without further purification.

### 4.1. Concentration Measurement

The concentrations of the starting and conjugated products were measured using a UV-Vis Spectrophotometer Lambda 25 (PerkinElmer) at 280 nm with quartz cells having an optical length of 1 cm. The theoretical molar extinction coefficient (ε_280_ = 1.917) was calculated using the ProtParam website (https://web.expasy.org/protparam/ accessed on 10 November 2023) considering the molecular weight and tryptophan, tyrosine and cysteine composition of scFvD2B.

### 4.2. HPLC Analysis Methods

High-performance liquid chromatography (HPLC) analyses were carried out using two methods:

SE-HPLC (method A) was performed in an Agilent instrument using ChemStation OpenLab Server software V2.7) with a quaternary pump coupled to a UV-photodiode array detector set at 226 nm interfaced with a sodium iodide radiometric detector (GABI Raytest) using a size-exclusion Bio Sec-5 column (300Å, 7.8 × 300 mm) (Agilent, Santa Clara, CA, USA) as stationary phase. The mobile phase was PBS (0.1 M, pH 7); 0–25 min, isocratic; the flow rate was 1 mL/min.

C_4_-RP-HPLC (method B) was conducted in a Dionex Ultimate 3000 (Thermo Fisher Scientific^TM^, Waltham, MA, USA) HPLC equipped with a UV detector set at 226 nm and a radioisotope detector (Gabi Raytest) using a reverse-phase Symmetry300 C4 column (5 μm; 4.6 × 150 mm). The mobile phase was solvent A: H_2_O milliQ with 0.1% (*v*/*v*) TFA and solvent B: CH_3_CN with 0.085% (*v*/*v*) TFA. The gradient step method was: 0 min, %B = 10; 3–11 min, %B = 50; 11–20 min, %B = 58; 20–21 min, %B = 95; 21–24 min, %B = 95; 24–25 min, %B = 10; the flow rate was 0.8 mL/min.

### 4.3. Conjugation of HYNIC-GGC-NH2 Peptide to scFvD2B-Tag (HscFv1)

All scFvD2B production batches were analyzed with HPLC using the system described above to verify their purity before starting the reaction. The scFvD2B-Tag conjugation was carried out by adding N-(3-Dimethylaminopro-pyl)-N′-ethylcarbodiimide hydrochloride (EDC; 11.5 mg) and N-hydroxysuccinimide sodium salt (S-NHS; 3.20 mg) to 0.4-mL scFvD2B-Tag solution (5 mg/mL) in carbonate buffer (0.2 M; pH 9.5). The reaction mixture was incubated at 25 °C, stirring for 4 h to activate the terminal carbonyl groups of scFvD2B-Tag. Then, 80 μL of HYNIC-GGC solution (3 mg/mL) in carbonate buffer (0.2 M; pH 9.5) was added, and the reaction mixture was incubated at 37 °C, stirring for 24 h. To remove the excess reagents, the product was purified with a Vivaspin^®^ (Sartorius, Gottingen, Germany) concentrator (MWCO 5 kDa), washed 3 times with PBS (0.1 M, pH 7) and centrifuged for 30 min at 3000× *g* and 10 °C.

Finally, the purified products were analyzed using SE-HPLC (method A) to confirm the conjugation.

### 4.4. MALDI-MS Characterization of HscFv1 Conjugate

MALDI/MS measurements were performed using an UltrafleXtremeTM MALDI-TOF/TOF instrument (Bruker Daltonics, Bremen, Germany) equipped with a 1-kHz smartbeam II laser (λ = 355 nm) operating in linear positive ion mode. The instrumental conditions were: IS1 = 25.00 kV; IS2 = 23.60 kV; lens = 6.50 kV; delay time = 250 ns. The matrix was sinapic acid (saturated solution in H_2_O/CH_3_CN with 0.1% (*v*/*v*) TFA). Five microliters of the sample were mixed with 5 µL of the matrix solution. One µL of the resulting mixture was deposited on a stainless-steel sample holder and allowed to dry before introduction to the mass spectrometer.

External mass calibration was performed using Protein Calibration Standard 2 (Bruker Daltonics, Bremen, Germany) and based on the average *m/z* values of Protein A [M+2H]^2+^ (*m/z* 22,307), Trypsinogen [M+H]^+^ (*m/z* 23,982), Bovine Serum Albumin [M+2H]^2+^ (*m/z* 33,216), Protein A [M+H]^+^ (*m/z* 44,613), and Bovine Serum Albumin [M+H]^+^ (*m/z* 66,432).

### 4.5. Conjugation of S-HYNIC to scFvD2B (HscFv2)

The native single chain (1 mg/mL) was concentrated with an Amicon^®^ Ultra-4 centrifugal filter (MWCO 10 kDa, Merk Millipore, MA, USA) in a Sorvall^TM^ ST16R centrifuge (Thermo Fisher Scientific^TM^, Waltham, MA, USA) at 6000× *g* for 14 min to obtain two solutions with different concentrations: 5 mg/mL and 2.2 mg/mL, respectively. To have the basic pH essential for the conjugation reaction, the buffer was changed from PBS to borate buffer (0.1 M, pH 8.5) using a PD SpinTrap^TM^ G25 (GE Healthcare Chicago, IL, USA).

The conjugation reaction was then carried out by adding a dropwise DMF solution of S-HYNIC to scFvD2B (18 µL, 23.3 mM for each mg of antibody fragment) to obtain a 1:20 or 1:10 molar ratio (scFvD2B/S-HYNIC) for 5 mg/mL and 2.2 mg/mL of scFvD2B solution, respectively. The reaction mixture was incubated, gently stirring for 5 h at room temperature, protected from light. Finally, the product was purified, and the buffer was replaced with PBS using overnight dialysis in a Slide-A-Lyzer G2 Dialysis Cassette (Thermo Fisher Scientific^TM^, Waltham, MA, USA) and then stored at −80 °C.

The concentration of the conjugated product was determined at 280 nm with UV-Vis spectroscopy, as described above.

### 4.6. LC-MS Characterization of HscFv2 Conjugate

To measure the mass of the HscFv2 conjugate, LC-MS was performed with a Xevo G2-S Q-Tof mass spectrometer (Waters, Milford, MO, USA) connected to an Acquity H-Class UPLC system (Waters). Separations were carried out on a BEH C4 column (300 Å, 1.7 µm, 1 mm × 50 mm, Waters) using mobile phase A composed of water and 0.1% formic acid and mobile phase B of 0.1% formic acid in CH_3_CN. The analyses were performed using a linear gradient from 10% to 70% of B in 4 min at a flow rate of 0.1 mL/min. The Xevo G2-S QTof was operated in the ESI positive ion resolution mode with the following source parameters: capillary 1.5 kV, sampling cone voltage 30 V, source offset of 80 V. MassLynx software V4.1 (Waters) achieved instrument control, data acquisition and processing.

### 4.7. Determination of the Conjugates’ Affinity for the PSMA

An ELISA assay was performed to evaluate the affinity and specificity of HscFv1 and HscFv2. Briefly, a Nunc Maxisorp 96-well plate was coated with recombinant PSMA (ACROBiosystems, Newark, DE, USA) or BSA (1 μg/100 μL, 50 μL/well) at 4 °C (overnight). Then, the wells were washed with 200 μL/well of washing buffer (0.05% Tween-20 in TBS, pH 7.4) four times and blocked with TBS/Tween + 4% nonfat dry milk, NFDM, (Sigma, St. Louis, MI, USA) at 37 °C for 1 h. Three washes were performed before incubating at 4 °C with serial dilution of HscFv1/HscFv2 or the parental scFvD2B. The plate was washed and incubated with Biotin-protein L (Thermo Fisher-Pierce, Waltham, MA, USA) 1:250 in TBS/Tween + 4% non-fat dry milk, NFDM at rt (room temperature) for 1 h. The plate was washed three times and incubated with Streptavidin-HRP 1:250 in TBS/Tween + 4% NFDM at rt for 1 h. Finally, the plate was washed and stained with TMB (Sigma), and the enzymatic detection was measured using a plate reader.

Displacing experiments were performed on LNCaP PSMA+ cells; briefly, the cells were incubated with a fixed amount of the parental whole Ab D2B (i.e., 0.1 ug) alone or mixed with different molar concentration of HscFv1 or scFvD2B-Tag (i.e., 500×–25×–1× molar excess) for 1 h at 4 °C. After three washes with PBS, the cells were stained with an anti-mouse IgG FITC-labeled secondary antibody (SIGMA, Milano, Italy) for 30 min at 4 °C. After a final wash in PBS, the cells were collected and analyzed using a BD FACSCanto II apparatus (BD Biosciences Franklin lakes, NJ, USA). Data on the percentage of cells gated in the FITC channel were collected.

### 4.8. Radiolabeling of HscFv1 and HscFv2 Conjugates

HscFv1 and HscFv2 were radiolabeled by adding 100 µL of the selected conjugate (9.02–2.28 nmol in PBS 0.1 M, pH 7) to a glass vial containing 75 µL of a 1:1 EDDA and tricine mixture (EDDA, 20 mg/mL of NaOH 0.1 M; Tricine, 40 mg/mL of phosphate buffer 0.2 M pH 6.2), followed by SnCl_2_ (10 µL; 1mg/mL of 0.1 N HCl;) and ^99m^TcO_4_^−^ (370 MBq/75 µL). The reaction mixture was incubated at 37 °C for 1 h. All products were analyzed at the end of the incubation time with HPLC, as described before, to assess the labeling efficiency of the reactions.

^99m^Tc-HscFv1 and ^99m^Tc-HscFv2 were purified with gel filtration using NAP-5 columns (GE Healthcare, Amersham, UK, cod. 17-0853-01) and 420 μL of PBS as elution volume. The products were then analyzed with SE-HPLC and C_4_-RP-HPLC to determine the percentage of radiochemical purity (%RCP) and utilized for stability and cellular studies.

### 4.9. Stability Studies of HscFv1 and ^99m^Tc-HscFv2

To assess the stability of both conjugates, they were diluted 1:10 in PBS (0.1 M; pH 7.0) or SS and incubated for 24 h at 37 °C. The human serum stability of ^99m^Tc-HscFv2 was also evaluated by incubating the compound diluted 1:10 in human serum under the same conditions. The percentage of RCP was then determined for all the samples using C_4_-RP-HPLC at different time points ranging from 15 min to 24 h.

### 4.10. Cell Binding and Internalization

To assess the cell uptake and internalization of ^99m^Tc-conjugates, the purified ^99m^Tc-HscFv1 and ^99m^Tc-HscFv2 (5 μCi/5 μL) were added to a suspension (1 × 10^6^ cells/1 mL of RPMI) of the selected PC3-PIP(++), LNCaP(+) or PC3(−) and incubated at 37° C for 1.5 h (in triplicate). At the end of the incubation time, the cells were centrifuged at 3000 rpm for 5 min at 4 °C to separate the supernatant (S1) from the pellet (P). The pellet was washed with PBS (1 mL) and centrifuged at 3000 rpm and the supernatant was collected and added to S1. To determine the fraction of ^99m^Tc-conjugate that was internalized by the cells, the membrane-bound radiotracer was eliminated by washing the cells with 1 mL of glycine buffer (pH 2.8) for 1 min. The cells were then centrifuged at 3000 rpm for 10 min at 4 °C and the supernatant (S2) was separated from the pellet. The radioactivity in each fraction was measured using a gamma-counter and the percentage of cell uptake and internalization was determined as follows:$$\%\;uptake=100-\frac{1}{n}\sum_{i=1}^{n}{\left(\frac{S1}{S1+S2+P}\times 100\right)}_{i}$$
$$\%\;internalization=\frac{1}{n}\sum_{i=1}^{n}{\left(\frac{P}{S1+S2+P}\times 100\right)}_{i}$$

To demonstrate the target specificity of the products, uptake and internalization tests were also performed simultaneously in the presence of fourfold (1.65 μmol) non-labeled scFvD2B as a blocking agent, which competes for the receptors. The experiment was repeated twice on different dates.

### 4.11. Statistical Analysis

All statistical analyses were performed using OriginPro 2016 (64-bit) Software (OriginLab Corporation). Differences in cancer cell uptake and internalization between the two agents and the unblocked and blocked receptors were evaluated using the two-sample t-test, assuming unequal variance (Welch correction), with a 95% confidence interval and two-tailed. The data distribution of each group compared was tested using the Shapiro–Wilk normality test to confirm that they were from a normally distributed population before performing the statistical *t*-test.

## 5. Conclusions

The HYNIC-scFvD2B conjugates produced and characterized in this study exhibit high in vitro stability and specific recognition for PSMA. In vitro cellular assays demonstrated that the internalization capacity of scFvD2B conjugates in PSMA+ cancer cells was >tenfold than that of HYNIC-conjugated peptides. However, the reduced labeling yield makes the use of ^99m^Tc-HscFv1 unfeasible; in contrast, the labeling yields obtained for ^99m^Tc-HscFv2 allow for further studies to optimize its production.

Nevertheless, the results of this investigation warrant further preclinical studies to determine whether the in vivo stability and tumor uptake of ^99m^Tc-HscFv2 still offer sufficient advantages over HYNIC-conjugated peptides to consider it a promising agent for SPECT/PSMA imaging. Meanwhile, considering the good properties of scFvD2B-Tag, it would be worth exploring more appropriate labeling methods.

## Data Availability

Data is contained within the article.

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
