# Peer review of "Development and Characterization of ^99m^Tc-scFvD2B as a Potential Radiopharmaceutical for SPECT Imaging of Prostate Cancer"

_ijms, 2023, doi:10.3390/ijms25010492_

Round 1
Reviewer 1 Report
Comments and Suggestions for Authors
My comments are mainly aimed at improving the clarity of the manuscript:
-The abstract should be written in a structured form, e.g. subdivided into Background/Purpose, Materials and Methods, Results and Conclusions sections. Moreover:
a) The first sentence (lines 20-23) should be removed, and the study background should eventually be put in a broader research context for better clarity.
b) The Results should be reported as much as possible in quantitative terms, along with the p-values of the statistical tests used to determine their statistical significance.
Main body. The composition should be thoroughly revised. First of all, the Materials and Methods should follow the Introduction and be followed by Results, Discussion and Conclusions.
-Introduction. It is rather lengthy and should be shortened by at least 30%, possibly moving some parts of it to the Discussion section.
-Line 496. The term "%internalizzazione" should be translated in English.
-Statistical analysis (lines 503-504). Which kind of t test? Paired t test? Two-tailed? Which level of statistical significance? Was the data distribution tested for normality prior to using the t-test? Was some kind of software used for statistical analysis, and eventually which?
Comments on the Quality of English LanguageMinor English language editing required
Reviewer 2 Report
Comments and Suggestions for Authors
In the article Development and characterization of 99mTc-scFvD2B as a potential radiopharmaceutical for SPECT imaging of prostate cancer the authors report the synthesis and characterization of a novel 99mTc labeled compound for prostate cancer imaging based on a IgG-D2B antibody fragment.
The authors clearly state the motivation to develop a diagnostic pair to the 177Lu variant of the compound based on the physical properties, availability and cost-effectiveness of the 99mTc radionuclide. Two compounds where synthesized and radiolabeled based on scFvD2B fragments, using HYNIC as chelating agent and were evaluated for affinity, specificity, radiochemical purity, stability and cellular uptake and internalization in prostate cancer cells. Both compounds show promising properties for PSMA imaging, demonstrating characteristics that are either better or comparable to those reported in current literature for HYNIC-conjugated peptides. However the authors report very low labeling yield for 99mTc-HscFv1, rendering it unsuitable for use in the clinical setting.
The authors note that scFvD2B can be conjugated to various chelating agents without significantly altering its binding affinity and pharmacokinetic properties, attributed to its larger size compared to PSMA-derived peptides. Future assessments of this 99mTc/177Lu–scFvD2B theranostics pair should explore how structural variances between the diagnostic and therapeutic agents affect their pharmacokinetic properties, as even minor differences can be relevant for dosimetry calculations for these compounds.
In conclusion, the work is well-written, the reported results are relevant to the development of radiopharmaceuticals and this manuscript deserves publication.
One minor error should be addressed. In line 496, the left hand term of the equation was not translated from Italian to English.
Author Response
We thank the reviewer for their helpful and constructive criticism, which has improved the quality of our work.
One minor error should be addressed. In line 496, the left-hand term of the equation was not translated from Italian to English.
The term “internalization” was correct and the corrections have been highlighted in the manuscript to assist the reviewer.
Round 2
Reviewer 1 Report
Comments and Suggestions for Authors
Thank you. No further comments.